# A Low Temperature Drifting Acoustic Wave Pressure Sensor with an Integrated Vacuum Cavity for Absolute Pressure Sensing

**DOI:** 10.3390/s20061788

**Published:** 2020-03-24

**Authors:** Tao Wang, Zhengjie Tang, Huamao Lin, Kun Zhan, Jiang Wan, Shihao Wu, Yuandong Gu, Wenbo Luo, Wanli Zhang

**Affiliations:** 1University of Electronic Science and Technology of China, School of Electronic Science and Engineering, Xiyuan Avenue, Chengdu 611731, China; tangzhengjie1996@163.com (Z.T.); zhankun1121@163.com (K.Z.); uestcjw@163.com (J.W.); luowb@uestc.edu.cn (W.L.); wlzhang@uestc.edu.cn (W.Z.); 2University of Electronic Science and Technology of China, State Key Laboratory of Electronic Thin Film and Integrated Devices, Xiyuan Avenue, Chengdu 611731, China; 3Shanghai Industrial μTechnology Research Institute, No. 235, Chengbei Road, Jiading District, Shanghai 201800, China; Huamao.Lin@sitrigroup.com (H.L.); Shihao.Wu@sitrigroup.com (S.W.); Alex.Gu@sitrigroup.com (Y.G.)

**Keywords:** piezoelectric, acoustic wave, pressure sensor, temperature compensation, vacuum cavity

## Abstract

In this paper we demonstrate a novel acoustic wave pressure sensor, based on an aluminum nitride (AlN) piezoelectric thin film. It contains an integrated vacuum cavity, which is micro-fabricated using a cavity silicon-on-insulator (SOI) wafer. This sensor can directly measure the absolute pressure without the help of an external package, and the vacuum cavity gives the sensor a very accurate reference pressure. Meanwhile, the presented pressure sensor is superior to previously reported acoustic wave pressure sensors in terms of the temperature drift. With the carefully designed dual temperature compensation structure, a very low temperature coefficient of frequency (TCF) is achieved. Experimental results show the sensor can measure the absolute pressure in the range of 0 to 0.4 MPa, while the temperature range is from 20 °C to 220 °C with a TCF of −14.4 ppm/°C. Such a TCF is only about half of that of previously reported works.

## 1. Introduction

Microelectromechanical system (MEMS) devices have been developed and widely tremendously in the past decades [1,2,3,4,5,6]. Pressure sensors are one of the most significant branches of MEMS devices in the commercial market. Devices that can measure absolute pressure changes are desired in many scenarios like the automotive or aerospace industry as well as fossil fuel detection due to the fact their readouts are invulnerable to environmental perturbations [7,8,9,10,11]. However, most MEMS pressure sensors can only sense relative pressure changes by measuring the pressure difference on the both side of a diaphragm. In order to measure the absolute pressure, the pressure sensor needs an extra vacuum package to achieve a direct absolute pressure readout [12,13,14]. This is a negative factor for minimizing the device size and might reduce device’s reliability as working time increases. 

For harsh environment applications, such as high temperature scenarios, the sensor is required to withstand high temperatures with low temperature drift. Unfortunately, conventional silicon-based piezoresistive pressure sensors easily lose their accuracy due to the change of resistivity and unstable metal/semiconductor contact. Such sensors can hardly work when the environmental temperature is higher than 150 °C. A promising solution is the use of acoustic wave technology, e.g. surface acoustic waves (SAWs) and Lamb waves. Acoustic wave devices have gained much more attention in the last few decades because of their low cost, small size, low power consumption and wireless operation [15,16,17,18,19,20]. However acoustic wave pressure sensors still suffer an unavoidable frequency shift vs. temperature, and this will result in inaccurate pressure readouts. Kropelnicki et al. presented a dual mode acoustic wave pressure sensor to cancel the effect caused by temperature [21,22], at the cost of a complex readout circuit in actual work. Besides, such a sensor is designed for relative pressure measurements, not absolute pressure ones.

This paper describes an aluminum nitride (AlN)-based acoustic wave absolute pressure sensor, containing an integrated vacuum cavity as pressure reference. This sensor can directly measure absolute pressure changes from 0 to 0.4 MPa, without the help of an external package. Benefitting from the device design, the sensor reveals low temperature drift features. It is able to operate at temperature ranges from 20 °C to 220 °C with a first order temperature coefficient of frequency (TCF) of −14.4 ppm/°C, which is only about half that in previously reported works [21,22]. We believe this work extends the capability boundary of MEMS pressure sensors.

## 2. Design and Simulation

The design of this acoustic wave pressure sensor starts from a quite common and rough acoustic wave pressure sensor. It is formed by a stack of Si/AlN/Mo and an open silicon backside etched hole which was used as a relative pressure reference. The top Moly layer was patterned as the IDT and Bragg reflectors. In order to implement the absolute pressure sensing function, a pressure sensor integrated as a vacuum cavity was put forward as an absolute pressure reference cavity in the design to replace the opened backside etched hole. The whose cross-sectional view is shown in Figure 1. 

A bottom Moly layer has been designed to confine the electric field in order to achieve a larger effective coupling coefficient $k_{\mathrm{eff}}^{2}$. The resonator’s 2D COMSOL Multiphysics simulation results of electric field distribution and admittance with and without the bottom metal Moly layer are shown in Figure 2 and Figure 3, respectively. It is indicated in Figure 2 that the electric field of the acoustic wave resonator with a bottom metal layer is better constrained compared to the one without a bottom metal layer. The $k_{\mathrm{eff}}^{2}$ is heavily enhanced from 0.3% to 0.8% which can be gathered from the simulated admittance differences in Figure 3.

The design to compensate for low temperature drift must be taken into consideration for a more accurate sensing result since the sensing mechanism is based on the phase velocity variation in acoustic wave propagation direction due to the change of temperature or pressure. In actual measurements, such a phase velocity variation can be detected by the change of resonance frequency. The relationship between resonant frequency *f_s_* and phase velocity *v_p_* of the generated acoustic waves is given by the following expression:(1)fs=υpλ
where *λ* is wavelength of the generated acoustic wave which is determined by the IDT electrodes’ periodicity. Besides the phase velocity of the acoustic wave is given by:(2)υp2=Eρ
where *E* is the material Young’s modulus and *ρ* is the material mass density. Since the Young’s modulus has a strongly relationship with temperature, the temperature dependence of the Young’s modulus can be deduced as follows:(3)$${E(T)=E(T}_{0})(1+TCE\cdot \Delta T)$$
where *T* refers to the measured temperature, *T_0_* is defined as the reference temperature and *ΔT = T – T_0_* is the relative temperature change, *TCE* corresponds to the first order temperature coefficient. The temperature dependence of a material of mass density *ρ* is given by:(4)$${\rho (T)=\rho (T}_{0}{)(1-(\alpha }_{11}{+\alpha }_{22}{+\alpha }_{33})\cdot \Delta T)$$
where *α_11_*, *α_22_* and *α_33_* are the thermal expansion coefficients for different lattice vectors. The wavelength λ which is also affected by thermal expansion as given by:(5)$${\lambda (T)=\lambda (T}_{0}{)(1+\alpha }_{11}\cdot \Delta T)$$
where *α_11_* is the thermal expansion coefficient for the wave propagation path. Here to simplify the calculations, the Equations (3), (4) and (5) are approximated by a first-order Taylor polynomial. The shift of phase velocity for generated acoustic wave due to the change of material Young’s modulus and density is given by:(6)$$\frac{\Delta v_{p}}{v_{p}}=(\sqrt{\frac{Ε+\Delta Ε}{\rho +\Delta \rho }}\;-\;\sqrt{\frac{Ε}{\rho }})/\sqrt{\frac{Ε}{\rho }}=\frac{\Delta Ε}{2Ε}\;-\;\frac{\Delta \rho }{2\rho }$$
where *ΔE* and *Δρ* are the relative change of Young’s modulus and density which are caused by temperature changes. Furthermore, the shift of resonance frequency can be deduced by: (7)$$\frac{{\Delta f}_{s}}{f_{s}}=(\frac{v_{p}+\Delta v_{p}}{\lambda +\Delta \lambda }-\;\frac{v_{p}}{\lambda })/\frac{v_{p}}{\lambda }=\frac{\Delta Ε}{2Ε}-\frac{\Delta \rho }{2\rho }-\;\frac{\Delta \lambda }{\lambda }$$
where *Δλ* is the relative change of wavelength due to the temperature change. From the above calculations, the generated acoustic wave resonant frequency shift due to the external temperature change called temperature coefficient of frequency (TCF) is given as follows:(8)$$\mathrm{TCF}=\frac{1}{f_{s}}\frac{\partial f_{s}\left(T,P\right)}{T}=\frac{1}{2E}\frac{\partial E(T,P)}{T}\;-\;\frac{1}{2\rho }\frac{\partial \rho (T,P)}{T}\;{-\;\alpha }_{11}=\frac{1}{2}{(\mathrm{TCE}-\alpha }_{11}{+\alpha }_{22}{+\alpha }_{33})$$

The most common temperature compensation method is to deposit a silicon oxide layer, because most of the materials have negative temperature coefficient of Youngs modulus (TCE) except for quite a few materials such as SiO2 which has a positive TCE. Different positions of the oxide layer in the stack can also have different results for temperature compensation. The finite element method employing the COMSOL Multiphysics software was used for simulations to verify the temperature compensation design. 

Figure 4 shows the 2D resonator models of one unit used in the simulation with periodic boundary conditions. The simulation models’ boundary conditions and materials constants are shown in Table 1 and Table 2 respectively. The curves of temperature dependence resonance frequency with different positions of oxide layer were simulated and plotted in Figure 5, where the oxide layer has been deposited in the bottom of the silicon layer, between the silicon layer and the AlN layer as well as on the top layer, respectively.

It can be found in Figure 5 that the lowest temperature coefficient (TCF) can be obtained when the oxide layer was inserted between the silicon and the AlN layer, but as is known, the oxide layer will decrease the device Q-value. Besides, the closer the oxide layer is to the wave energy center, the more tremendous a decrease of Q-value and resonance frequency will be found. In order to get lower TCF in a much reasonable way without harming the device quality, double temperature compensation layers have been designed on both the bottom and top of the stack. The bottom and top oxide layer were set to be 1 μm and 0.7 μm, respectively. Figure 6 shows the simulated double temperature compensated layers of the acoustic wave resonant mode shape and its temperature dependence on the resonance frequency relationship. A much lower temperature coefficient of frequency was achieved to be TCF = −2.92 ppm/°C. The pressure sensor with double oxide temperature compensation layers is supposed to have an excellent temperature compensation capacity according to the simulation result in Figure 6b.

From the above simulations, the optimized design of Lamb wave pressure sensor was put forward. Figure 7 demonstrate the 3D illustration of the Lamb wave pressure sensor and its cross-view. This Lamb wave pressure sensor carries an integrated vacuum cavity for directly absolute pressure sensing and has low temperature drifting. A stack of 0.02 μm AlN/0.2 μm Mo/1 μm AlN/ 0.2 μm Mo is deposited on an 8” CSOI wafer, which consists of a 5 μm device Si layer and 1 μm buried oxide (BOX) layer. The vacuum cavity used as a pressure reference cavity has a depth of 20 μm. A 0.7 μm SiO_2_ layer was then deposited on the top as a temperature compensation and passivation layer. The device for experiment tests has 40 pairs of IDT electrodes and 60 Bragg reflectors. The IDTs were designed to be double-electrode-type in order to eliminate the wave reflection in IDTs, each electrode has a line width of 1.25 μm, electrode space of 1.25 μm and length of 450 μm. The metal grate of the Bragg reflects has a line width of 2.5 μm, grate space of 2.5 μm and length of 455 μm.

## 3. Fabrication Process

A cavity SOI(CSOI) wafer was required as a substrate for the pressure sensor. The fabrication process of CSOI wafer was shown in Figure 8. First, a 20 μm deep cavity was patterned and etched by deep reactive ion etching (DRIE) in the base silicon wafer, followed by the backside alignment mark formation. On the other hand, the device silicon wafer was thermal oxidized to form a 1 μm SiO_2_ layer as a temperature compensation layer. After that, the base silicon wafer and the device silicon wafer were both polished on the bonding surfaces using chemical mechanic polishing (CMP) as a pre-processing for the following bonding process. Then these two wafers were bonded together using hydrophilic wafer-bonding technology, operated in a vacuum environment to ensure the success of bonding and to form the vacuum cavity. By the above process steps, a vacuum cavity was fabricated successful. Then the device wafer was thinned down to 5 μm silicon thickness from the top. The CSOI wafer with a 20 μm vacuum cavity as well as a 1 μm buried oxide (BOX) layer was fabricated successfully. Figure 9 shows a cross-view SEM image of the CSOI wafer. 

The fabrication process flow of the Lamb wave pressure sensor is shown in Figure 10. First a stack of 0.2 μm Mo/0.02 μm AlN/1 μm AlN/0.2 μm Mo is deposited on the top of CSOI wafer by using physical vapor deposition (PVD). The bottom thin AlN layer is using as a seed layer. After that a 0.2 μm SiO_2_ layer is patterned and deposited by plasma enhanced chemical vapor deposition (PECVD) as a hard mask of the top Moly and the top Moly is patterned and etched using a reactive ion etch (RIE) to define the IDTs and Bragg reflectors’ structure on top of the piezoelectric layer to excite the acoustic wave. Next, a 0.7 μm layer of SiO_2_ is deposited and patterned. This oxide layer is going to be used as an inner layer dielectric to isolate the aluminum connection line and top Moly. Then, the bottom and top Moly electrodes contact is opened connecting the top Al pad. Finally, the top Al connection was patterned and deposited to connect the top and bottom Moly electrodes. The device was fabricated in SITRI using its 8 in MEMS fabrication factory. Figure 11 shows the XRD curve of the AlN film and it demonstrates a very good AlN film quality with small full width at half maximums (FWHM) which is FWHM = 1.272°. Figure 12 shows the microscope image of the Lamb wave pressure sensor. 

## 4. Experiments and Results

The experiment was operated to measure the temperature and pressure coefficient of frequency. The short-open-line-thru (SOLT) method was used to calibrate the vector network analyzer (VNA). The schematic of the testing system was shown in Figure 13. The sensor chip was connected to the printed circuit board (PCB) using wire bonding.

For the temperature and pressure character testing, the sensor was put into an oven and a metal chamber for testing, respectively. The applied temperature and pressure were controlled by the temperature and pressure controller, respectively. The temperature dependent S11 parameters and the temperature dependent resonance frequency were plotted in Figure 14. Meanwhile, the first-order temperature coefficient of frequency was extracted to be TCF= –14.4 ppm/°C from Figure 14b, which indicates a reasonable consistency comparing to the simulation results in Figure 6b. In order to determine the pressure coefficient of frequency (PCF) character, a range of external pressures from 0 to 0.4 MPa was applied on the surface of the acoustic wave pressure sensor. Figure 15 describes the curves of the S11 parameters with different pressure conditions at room temperature and the pressure dependent resonance frequency. As is indicated in Figure 15a, the Q-value of the device was found to be over 10000 without any applied pressure. In the meantime, the first-order pressure coefficient of frequency (PCF) is extracted to be about PCF= 2.65 ppm/KPa. This large PCF value was got due to a very thin stack thickness of the device which is around 9 μm. The thin stack membrane also provided a good resolution. It was capable to distinguish a pressure change about 10 KPa which is the minimum range of the pressure controller, so its resolution may be far more than 10 KPa. As is mentioned above, this pressure sensor is also able to measure absolute pressure changes because a vacuum cavity was fabricated as a pressure reference system.

In addition, the sensor barely affected by external environment conditions due to the cavity was integrated into the sensor chip.

## 5. Conclusions

In this paper, an AIN-based Lamb wave pressure resonator with an integrated vacuum cavity is studied. The design, fabrication process, simulation and experiment results have been discussed above, respectively. The resonance frequency and Q-value of device was calculated to be 819.5 MHz and over 10000 at room temperature without any external pressure applied to the sensor. This pressure sensor is able to directly measure external pressure changes because of the integrated reference vacuum cavity. The die and function area size of sensor chip is 1500 × 1200 μm^2^ and 1000 × 300 μm^2^, respectively. The temperature coefficient of frequency (TCF) was measured to be −14.4 ppm/°C and the pressure coefficient of frequency was measured to be PCF = 2.65 ppm/KPa with excellent resolution, respectively. At the same time, the insertion loss of the device is up to 3 dB, which doesn’t reach the expectation. Further analysis was needed to decrease the insertion loss for a better device performance. The pressure ceiling hasn’t been reached to make sure its measurement range due to the equipment limits. This will be figured out to in later study. 

## Figures and Tables

**Figure 1 sensors-20-01788-f001:**
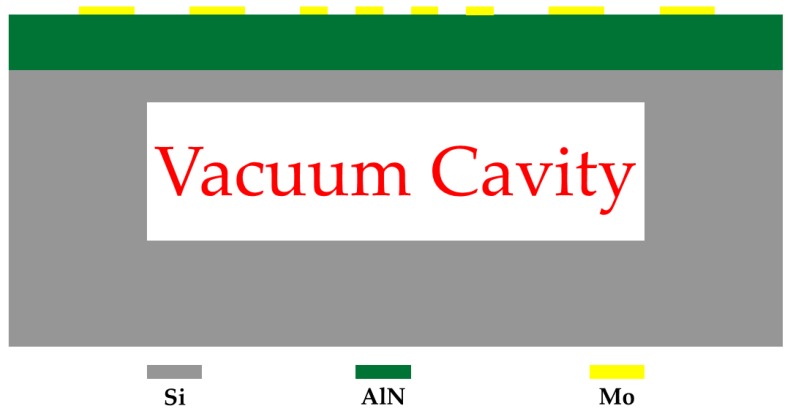
The cross-sectional view of a refreshed acoustic wave pressure sensor design with a deep vacuum cavity.

**Figure 2 sensors-20-01788-f002:**
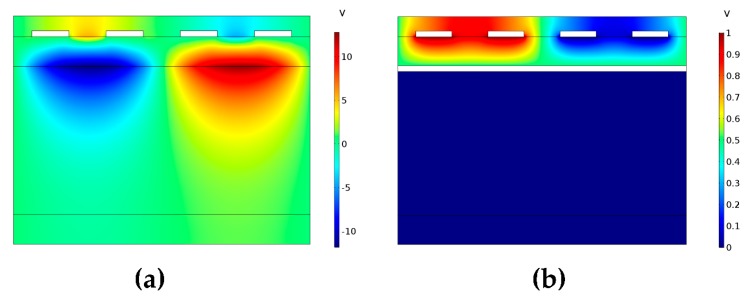
(**a**) The simulated electric field distribution of acoustic wave resonator without bottom metal Moly layer, (**b**) the simulated electric field distribution of acoustic wave resonator with bottom metal Moly layer.

**Figure 3 sensors-20-01788-f003:**
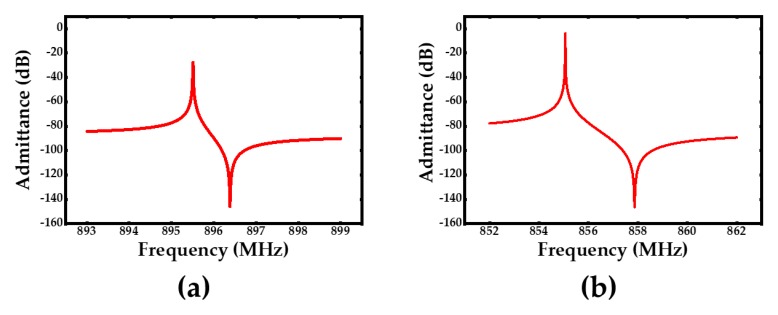
(**a**) The simulated admittance of acoustic wave resonator without bottom metal Moly layer, (**b**) the simulated admittance of acoustic wave resonator with bottom metal Moly layer.

**Figure 4 sensors-20-01788-f004:**
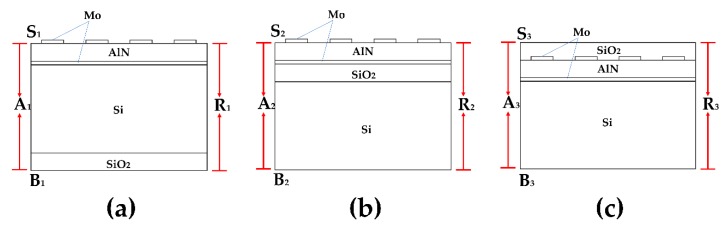
One unit of 2D resonator simulation models with periodic boundary conditions, (**a**) the oxide layer in the bottom of Silicon layer, (**b**) the oxide layer between the Silicon and AlN layer, (**c**) the oxide layer on the top layer.

**Figure 5 sensors-20-01788-f005:**
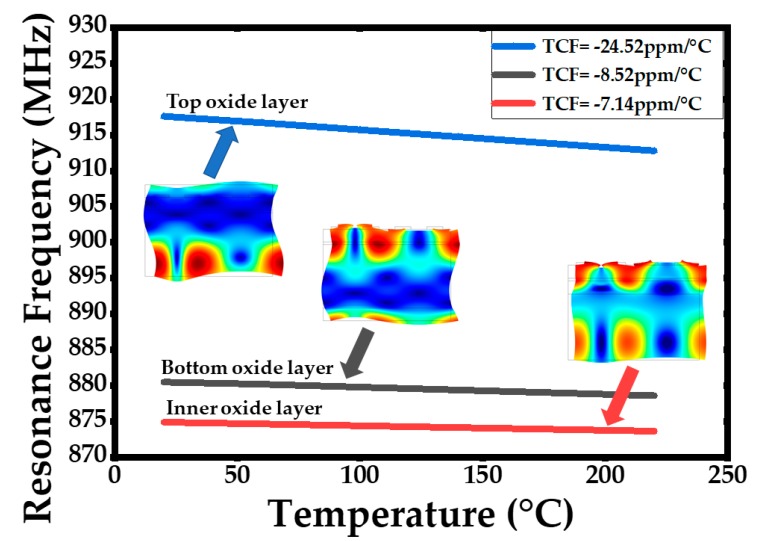
The simulated temperature dependent resonance frequency of acoustic wave resonator with different oxide layer position.

**Figure 6 sensors-20-01788-f006:**
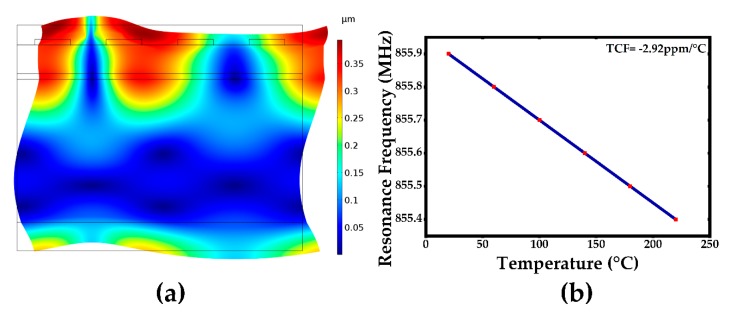
(**a**) The simulated acoustic wave mode shape (resulting displacement) with double temperature compensation layer, (**b**) temperature dependent of resonance frequency of acoustic wave resonator.

**Figure 7 sensors-20-01788-f007:**
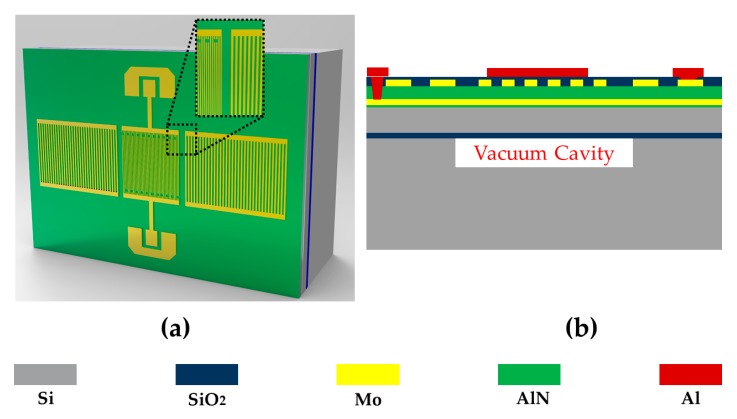
(**a**) The 3-D illustration of the acoustic wave pressure sensor; (**b**) cross-sectional view of the illustration.

**Figure 8 sensors-20-01788-f008:**
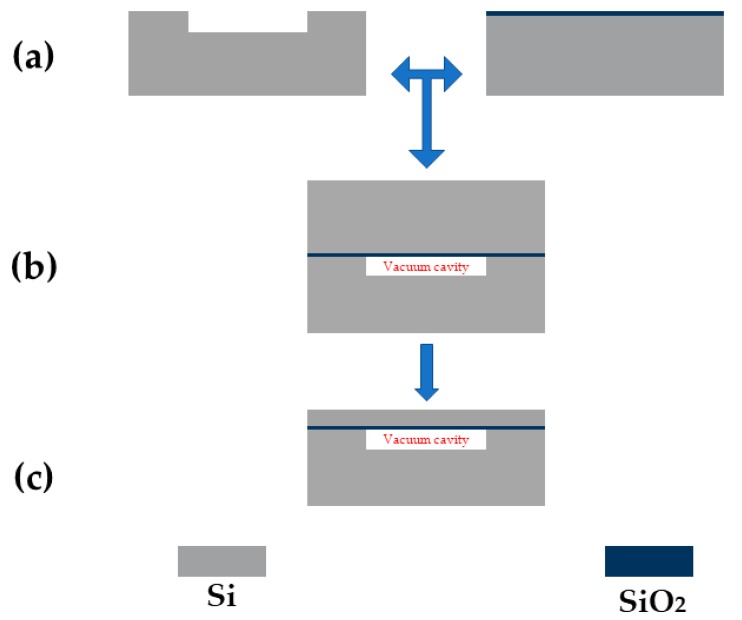
Fabrication process flow of the CSOI: (**a**) a 20 μm deep cavity are patterned and etched by deep reactive ion etch (DRIE) in the base silicon wafer, in the meantime, the device silicon wafer was thermal oxidized to form a 1 μm oxide layer on the surface as a temperature compensation layer, (**b**) the handle wafer and the device wafer were bonding together using hydrophilic bonding technology, (**c**) thinning the device wafer down to 6 μm which contains 1 μm SiO2 layer and 5 μm silicon layer.

**Figure 9 sensors-20-01788-f009:**
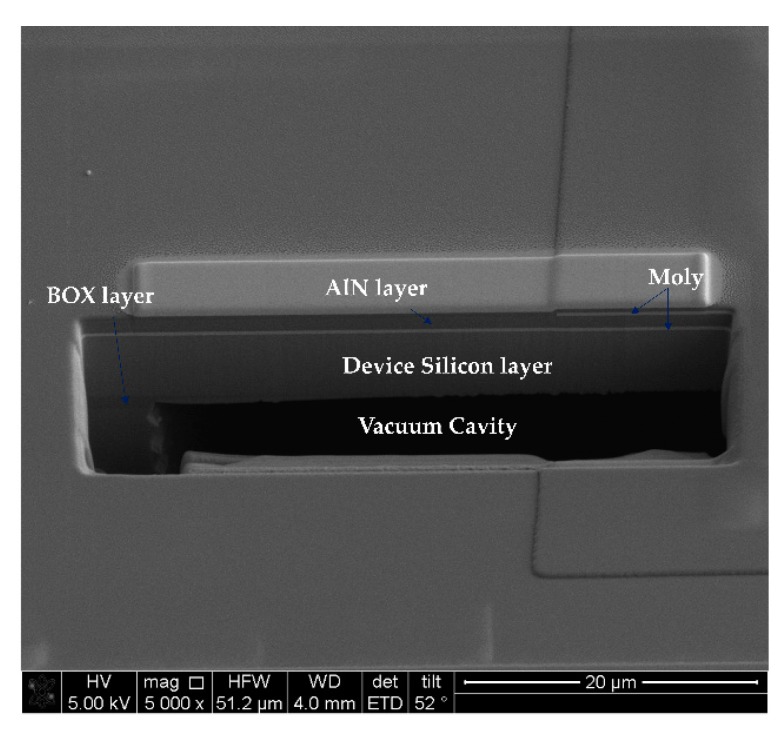
The cross-view SEM image of the device substrate with a vacuum cavity.

**Figure 10 sensors-20-01788-f010:**
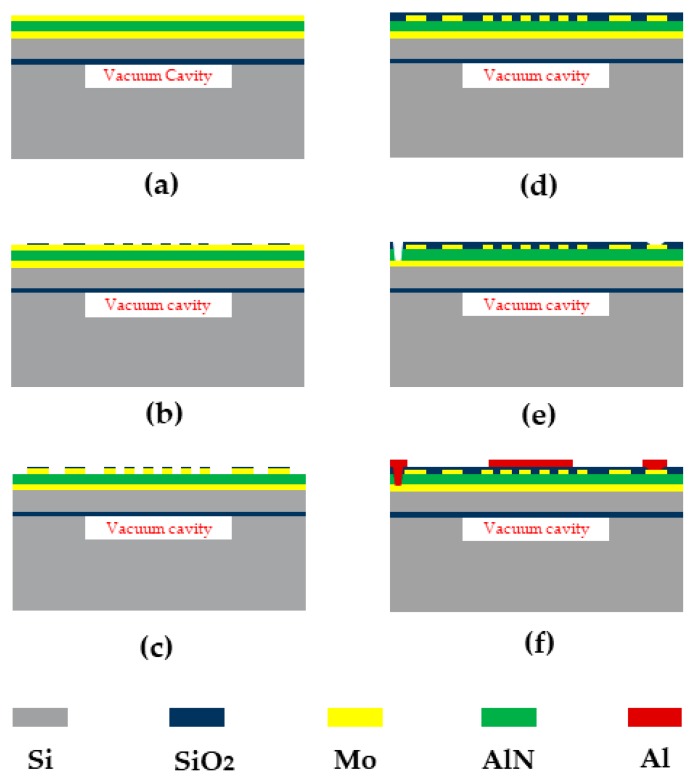
Fabrication process flow of the acoustic wave pressure sensor: (**a**) a stack of 0.2 μm Mo/0.02 μm AlN/1μm AlN/0.2 μm Mo is deposited by PVD; (**b**) a 0.2 μm SiO_2_ layer was patterned and etched as a hard mask of the IDT and Bragg reflectors; (**c**) the top Moly is patterned and etched for IDTs and Bragg reflectors structures; (**d**) a 0.7 μm layer of SiO_2_ is deposited and patterned as a temperature compensation layer and an inner layer dielectric to isolate the aluminum connection line and top Moly; (**e**) The bottom and top Moly electrodes contact is opened by DRIE to connecting the Al pad; (**f**) the top Al connection was patterned and deposited to connect the top and bottom Moly electrodes.

**Figure 11 sensors-20-01788-f011:**
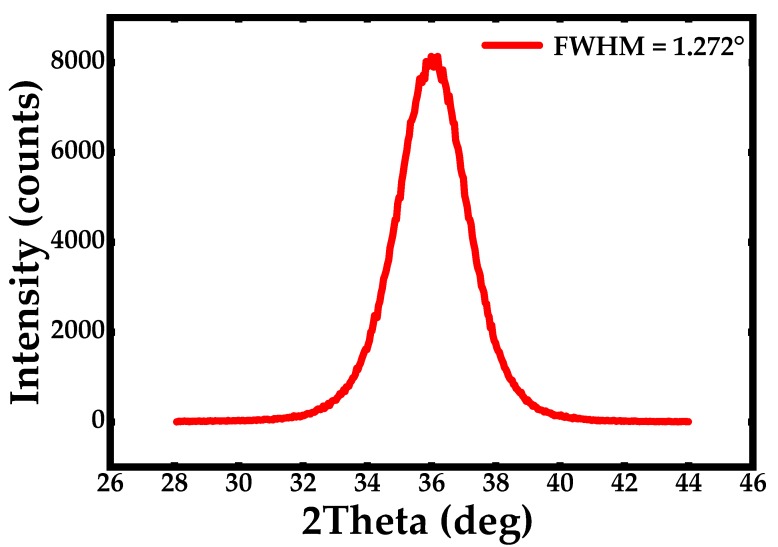
The XRD curve of the AlN film with FWHM = 1.272°.

**Figure 12 sensors-20-01788-f012:**
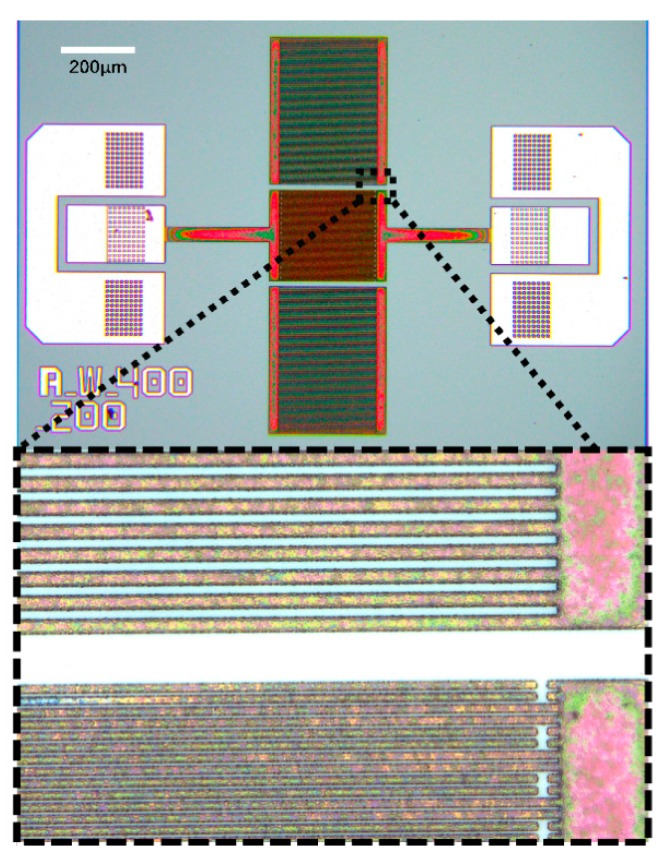
Microscope image and cross-view SEM image of the Lamb wave pressure sensor.

**Figure 13 sensors-20-01788-f013:**
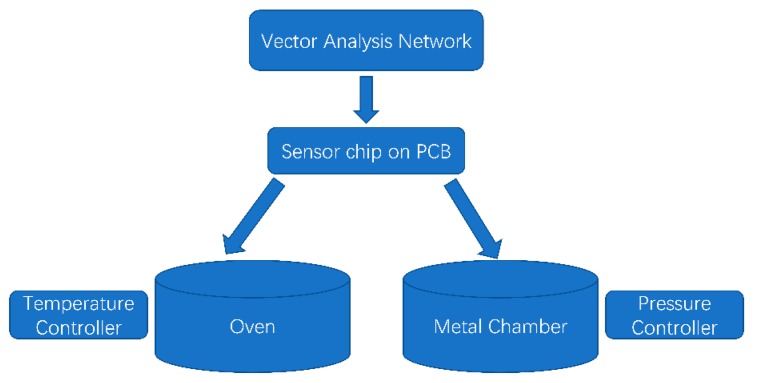
The schematic of the Lamb wave pressure sensor temperature and pressure character testing system.

**Figure 14 sensors-20-01788-f014:**
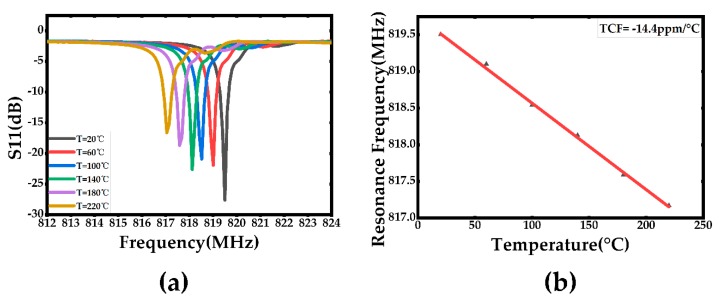
(**a**) The temperature dependent S11 parameters with different temperature, (**b**) the temperature dependent resonance frequency with different temperature.

**Figure 15 sensors-20-01788-f015:**
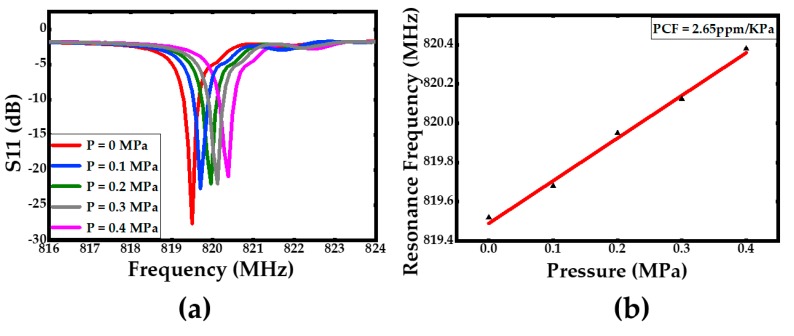
(**a**) The Lamb wave resonator S11 parameters with different pressure in room temperature, (**b**) the pressure dependent resonance frequency with different pressure.

**Table 1 sensors-20-01788-t001:** Boundary conditions of the 2D simulation model.

Boundary	Mechanical Boundary Conditions Setting
S1, S2, S3, B1, B2, B3	Stress-free boundary
A1, A2, A3, R1, R2, R3	Periodic boundary

**Table 2 sensors-20-01788-t002:** Materials constants used in the simulation.

	AlN			Si	SiO_2_	Mo
Elastic constants, *c_ij_* (GPa)	*c_11_* *c_12_* *c_13_* *c_33_* *c_44_* *c_66_*	410.06 100.69 83.82 286.24 100.58 154.70	Young’s modulus *E* (GPa)	170	70	385
Temperature coefficient of elastic constants *TEC_ij_* (10e-6/K^−1^)	*TCE_11_* *TCE_12_* *TCE_13_* *TCE_33_* *TCE_44_* *TCE_66_*	−10.65 −11.67 −11.22 −11.13 −10.82 −10.80	Temperature coefficient of Young’s modulus *TCE* (10e-6/K^−1^)	−63	200	−181
Thermal expansion*α_ij_* (10e-6/ *K^−1^*)	*α_11_* *α_22_* *α_33_*	5.27 5.27 4.15		2.6 2.6 2.6	0.55 0.55 0.55	3.49 3.49 3.49
Mass density *ρ*(kg/m^3^)	*ρ*	3300		2329	2200	10,200

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
