# Peer review of "A Low Temperature Drifting Acoustic Wave Pressure Sensor with an Integrated Vacuum Cavity for Absolute Pressure Sensing"

_sensors, 2020, doi:10.3390/s20061788_

Round 1
Reviewer 1 Report
The authors proposed a low temperature drifting SAW pressure sensor. The two main points are using SiO2 as temperature compensation layer and integration of a vacuum cavity for absolute pressure measurement. But these methods are lack of novelty. Additionally, the manuscript need major revisions to improve the quality.
The manufactured SAW sensor are two-port device, it should use S21, not S11 only, to characterize the devices. Please provide both the S21 and S11 in Fig. 13 and Fig. 14. In Fig. 13a, the authors provide S11 at different temperatures and deduce the TCF as Fig. 13b. But in Fig. 14a, the authors only provide one curve at one specified pressure. The pressure sensing characteristics are the main considerations of this paper. Please provide the S21 and S11 under different pressure conditions. Please describe the design parameters of SAW devices, such the size of IDT, refections, and the gap between IDT and reflections. As shown in Fig. 11, the authors grouped two parallel electrodes as on electrode of conventional IDT. Why use this design? What is the pros and cons of this design? Please provide detail information. In Fig. 5, the gray line (top oxide layer) has smallest TCF. But in line 118, the authors say: "the lowest TCF can be gathered when the oxide layer was insert between the silicon and AlN layer" In Fig. 5, the displacement plots are not consistent and not related to the Lamb mode. The AlN film should be c-axis oriented, please provide the deposition parameters and measured XRD curve. In Fig. 10, 20nm AlN film is used as the seed layer before deposition of the Mo electrode. But conventionally, the seed layer is between the Mo and AlN piezoelectric layer. There should be bulk wave modes if a bottom Mo electrode is used. Please describe it. In Comsol simulation, the TCE data of Mo should be included. In line 200, Fig. 9c should be Fig. 6bAuthor Response
Please see the attachment.

Reviewer 2 Report
Reviewer’s report
Title: A low temperature drifting acoustic wave pressure sensor with an integrated vacuum cavity for absolute pressure sensing
Summary:
The authors have developed a novel acoustic wave pressure sensor based on AIN thin film with integrating vacuum cavity. It can measure absolute pressure due to the vacuum cavity to provide accurate reference pressure. Additionally, it was superior for temperature drifting (low TCF) because of the dual temperature compensation structure.
Design, simulation, and experimental results were explained with too many illustrations, However, there was a lack of explanation and analysis of important experimental results.
In figure 1, 7, 8, and 10, mark the cavity region.
In figure 3, match the range of y-axis between (a) and (b)
You mentioned, “It can be found in Figure 5 that the lowest temperature coefficient (TCF) can be gathered when the oxide layer was insert between the silicon and AlN layer.” About explained figure 5. Please check again. The resonance frequency was the lowest in the inner oxide layer case. However, it’s not the lowest TCF.
There were many problems with English expressions, including singular and plural errors.
“was insert”? à inserted
“according to the simulate result” à simulated
And so on.
Round 2
Reviewer 1 Report
The authors revised the manuscript carefully and provided required data. I think the paper can be accepted after following minor revisions:
- There is a typo in caption of Fig. 3. The (b) should be "with bottom metal moly layer", not "without".
- In XRD pattern, it is common to use $2\theta$ as the x-axis instead of $\theta$, and the (002) orientation of AlN thin film will around 36 degree. Please modify the x-axis of Fig. 11.
